**Data Availability Statement:** All relevant data are within the manuscript and its Supporting information files.

**Funding:** The project was financially supported by Ethiopian Institute of Agricultural Research. The

# Vegetables contamination by heavy metals and associated health risk to the population in Koka area of central Ethiopia

**Leta Danno Bayissa** [ID]*, **Hailu Reta Gebeyehu**¤

Department of Chemistry, College of Natural and Computational Sciences, Ambo University, Ambo, Ethiopia

¤ Current address: Ethiopian Institute of Agricultural Research, Addis Ababa, Ethiopia
* bayissa.leta@ambou.edu.et

## Abstract

Contaminated soil and vegetables have continued to instigate threat to human health globally and specially in developing countries. This study was aimed to determine concentrations of certain heavy metals in soil and vegetables (cabbage and tomato) from Koka area of central Ethiopia using Inductively Coupled Plasma Optical Emission Spectrophotometer (ICP-OES). The amounts of As, Pb, Cd, Zn, Cu, Hg and Co detected in soil samples were found to surpass the reference values for agricultural soil. Similarly, the concentrations of As, Pb, Cd, Cr and Hg obtained in both tomato and cabbage samples have exceeded the recommended values with the mean levels generally ranging from 0.93–6.76, 1.80–7.26, 0.33–1.03, 0.86–5.16 and 3.23–4.36 mg/kg dry weight, respectively. The result obtained have signified that leafy vegetable has hoarded heavy metals more than non-leafy vegetable. The total hazard quote for As and Hg from tomato ingestion and for As, Hg and Co from cabbage ingestion were greater than unity, signifying potential health hazard to the public. The health index (HI) owing to tomato and cabbage ingesting were 5.44 and 14.21, respectively, signifying likely adversative health implication to the population from the ingestion of the vegetables. The Total Cancer Risk (TCR) analysis have uncovered the possible cancer hazard persuaded by Cd, Hg, As and Ni from the ingestion of both vegetables. From the outcomes this study, it can be concluded that the soil and vegetables from Koka areas are possibly contaminated with toxic metals and hence demand strict monitoring to safeguard the public around the study area and beyond.

## Introduction

The quest for industrialization and the concern on food safety are ever increasing in both developed and developing nations all over the world. The concern about environmental pollution and food safety is growing due to the probable health danger to the population [1–3]. Among environmental contaminants, heavy metals due to the anthropogenic and other activities have attracted numerous attentions due to their serious health implication to humans when accrued in an preeminent concentration above body requirements [4, 5]. As evidenced

funders had no role in study design, data collection and analysis, decision to publish, or preparation of the manuscript.

**Competing interests:** The authors have declared that no competing interests exist.

by literature, vegetables demand and consumption is remarkably growing in every parts of the globe as it institutes a vital part of the human diet and nutrition [6, 7]. However, it has been documented that most of the vegetables commercially available, especially in developing nations, are often grown in urban and suburb areas of big cities [3, 8–10]. As a result, these vegetable are exposed to anthropogenic pollution instigated from sources including but not limited to urban and industrial wastes, mining and smelting and metallurgical industries [11, 12]. This evidently signifies that problems related to food safety and associated potential danger to the public have been a major apprehension all over the world [1, 13–15].

The pollution of soil with toxic heavy metals have been regarded as a foremost source of crops and vegetables contamination and a major route of human exposure to these toxic metals [16]. As a result, heavy metals have been recognized as substantial pollutants of vegetables cultivated around urban and suburb areas worldwide [3, 4, 11, 17–21]. This is a clear suggestion that vegetables being cultivated around urban and suburban areas are suggestively exposed to heavy metals contamination. It has also been signified that the contamination level and effect are more pronounced in developing nations compared with the developed one [5, 22].

Ethiopia, as a fastest growing country in sub-Saharan African countries, have attracted and being attracting many investors from all over the world. As a result, small and medium scale industries (mainly working on production of brews, fabrics, chemicals, floriculture and tanneries) are growing in a fastest rate and are generally established around urban and suburb areas and sideways of rivers [19, 23, 24]. The wastewater being released from these businesses are reportedly encompasses elevated levels of toxic metals including cadmium (Cd), arsenic (As), mercury (Hg), copper (Cu) and lead (Pb) [22, 25]. These toxic metals have been long regarded as serous environmental contaminants even at smaller concentration because of their detrimental effect to public health [1, 7, 15].

The buildup of toxic heavy metals in soil, water and plants due to hasty industrial expansion have been and still continued to be a great concern owning to their detrimental health implication to the public. However, the public health complication due to toxic heavy metals contamination has been evidenced to be more pronounced in developing nations compared with the developed ones [26]. The rapid industrialization and urbanization in Ethiopia has contributed and being contributing to the major environmental concern [10, 12, 19, 27, 28]. This clearly shows that, owing to the development, the environment and food safety is in jeopardy as the industries are releasing their wastes indiscriminately to the environment without or with little treatment [23, 29]. To this regard, in developing countries like Ethiopia, the environmental problems and potential health hazard to the population is more pronounced as the environmental safety regulations are not practically in place. Hence, this study was intended to evaluate heavy metal concentrations in cabbage, tomato and soil samples from Koka (Koka Ejersa and Koka Negewo) area farmlands in central Ethiopia. In addition, the probable health hazard related with the consumption of these vegetables have also been appraised through the determination of estimated daily intake (EDI), target hazard quotient (THQ), hazard index (HI) and target cancer risk (TCR) for selected heavy metals.

## Methodology

### Location map of the study areas

This investigation was focused on Koka areas (Koka Ejersa and Koka Negewo) farmlands which are situated at about 53 mils (85 km) southeast of Addis Ababa (the capital of Ethiopia) in Oromia Regional State. The location map of the area is shown in Fig 1. The area has a grid reference 8˚27.154′ latitude and 39˚03.894′ longitude with an average elevation of 1630 masl.

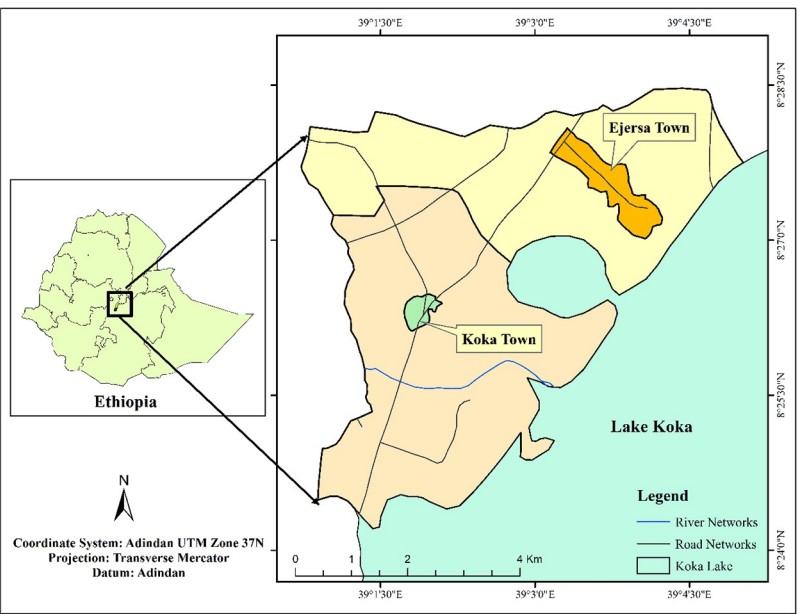

**Fig 1. Descriptive map of the study area (own drawing).**

Lake Koka (commonly known as Koka reservoir) is situated in Koka area around which small to medium sized industries including leather, textile, plastic and soap factories are located. The water bodies in Koka area, including Lake Koka, are highly susceptible to contamination due to the weak practices of waste disposal and management activities by the manufacturing companies positioned in the area. It has been witnessed during sample collection that majorities of manufacturing companies in the area are directly releases the wastes they have generated to the lake and/or to the rivers feeding the lake without or with little treatment.

## Vegetables and soil samples collection and preparation

The fresh cabbage (*Brassica oleracea*) and tomato (*Lycopersicon esculentum Miller*) were sampled using cleaned and decontaminated polyethylene bags. Both vegetables were separately sampled from five arbitrarily designated subsampling locations and mixed together to make up a 1kg each composite sample. Accordingly, a total of two cabbage and two tomato samples (one each from Koka Negewo and Koka Ejersa) were collected. All the vegetables samples were collected from carefully identified farmlands with the consent of the farmers. The collected samples were then directly taken to laboratory for subsequent handling and investigation. The sample treatment and preparation were done following similar procedure we have recently reported [19].

Soil sampling were performed from the same spots vegetable samples were collected. Generally, about 1 kg of soil samples were collected using polyethylene bags from the two sites (Koka Negewo and Koka Ejersa) exactly from the same spot vegetables were sampled (for each vegetable type separately) at depth of 0–20 cm by making use of a stainless steel auger soil sampler. Accordingly, a total of four soil samples (two soil samples on which tomato was grown and similarly two soil samples on which cabbage was grown from each site) were collected. The collected soil samples were packed with care, labeled and then taken to laboratory for treatment and examination as per the procedure we have recently reported [19].

## Methods validation procedures

The method detection limit (MDL) and limit of quantification (LOQ) have been adopted from similar study we have recent reported [19] and the data is presented in S1 Table in S1 File. Similarly, the recovery analysis procedures followed to attest method precision and accuracy were also adopted from our similar study result and the data were presented in S2-S4 Tables in S1 File. From the data, the spike recovery found falls within the general suitable assortment of 80–120% for a good recovery and authenticates the accurateness and reliability of the method employed for the analysis of metal levels. The lower percentage relative standard deviation (% RSD) values ($<$ 12%) found also showed that the method used was precise enough for the investigation of the heavy metals in the samples.

## Digestion procedures and metals analysis

A microwave digestion system was employed to digest both soil and vegetable samples by employing similar procedure we have recently reported [19] with minor modification as per the optimium conditions established (S5 Table in S1 File). From the optimization procedures, a mixture of $HNO_3$ (69%) and HCl (37%) acids at a volume ration of 9:3 mL was stablished at digestion time and temperature of 45 minutes 180 ˚C, respectively, to be optimum condition. These conditions were used to digest 0.5g of each soil and vegetable samples. The solutions obtained after digestion were filtered using Whatman No. 42 filter paper into 50 mL volumetric flask and its volumes were adjusted to the mark by dilute (2%) nitric acid. The levels of metals in the prepared soil and vegetables samples were determined by inductively coupled plasma optical emission spectrometry (ICP-OES) (Model: ARCOS FHS12, USA) following proper calibration and instrumental parameters setup as indicated in SM Table 6. The data obtained for each vegetable and soil samples were reported as mean ± SD from triplicate measurements.

## Estimation of bioconcentration factor (BCF)

The bioconcentration factor (BCF) as defined by Liu and co-workers [30] is the ratio of concentration of metal in comestible part of the plant to the metal concentration in soil sample. It has been described that, if BCF $\leq$ 1, the plant can only absorb the metal but not accumulate it. However, if BCF $>$ 1, the plant has been regarded as the likely accumulator of the metal [30, 31]. Consequently, the transfer extent of toxic metal from soil to plant was estimated by Eq (1) described by Sulaiman and Hamzah [31].

$$\text{BCF} = \frac{C_{plant}}{C_{soil}} \tag{1}$$

where $C_{plant}$ = heavy metal concentration in comestible portion of the plant; $C_{soil}$ = heavy metal concentration in the soil on which the plant is grown.

## Health hazard assessments

**Estimated daily intake (EDI).** Founded on the average levels of metals in individual cabbage and tomato and the appraised regular ingestion of the vegetables in gram, the estimated daily intake of the heavy metals measured in this investigation were determined. The EDI of individual metal considered in the study was estimated using Eq (2) as expressed by Chen and co-workers [32] with minor amendment.

$$\text{EDI} = \frac{E_f \text{ x } E_D \text{ x } F_{IR} \text{ x } C_M \text{ x } C_f}{B_W \text{ x } T_A} \text{ x } 0.001 \tag{2}$$

where $E_f$ = exposure rate (365 day/year); $E_D$ = exposure period (65 years), equivalent to average life time [33]; $F_{IR}$ = average vegetable consumption (240 g/person/day for low fruit and vegetable intake [34]); $C_M$ = metal concentration (mg/kg dry weight); $C_f$ = 0.085 (concentration conversion factor for fresh to dry vegetable weight) [35, 36]; $B_W$ = 70 kg (reference body weight for an adult) [29]; $T_A$ = average exposure time (65yrs x 365 days) and 0.001 = unit conversion factor. The complete information used for EDI estimation is presented in S6 Table in S1 File.

**Target hazard quotient (THQ).**   The non-cancer causing health hazard to the population around Koka area due to the ingestion of vegetables possibly polluted by toxic metals were appraised by the calculation of the target hazard quotient (THQ) using employing Eq (3) as defined by Chen and co-researchers [37].

$$\text{THQ} = \frac{\text{EDI}}{\text{RfD}} \tag{3}$$

Where EDI (mg/day/kg body weight) = estimated daily metal intake of the population and RfD = oral reference dose (mg/kg/day) values for all metals determined as presented in S6 Table in S1 File. It is generally regarded as safe from the risk of noncarcinogenic effects if the value of THQ is $< 1$. However, it is generally presumed that there is a possibility of noncarcinogenic effects if THQ is $> 1$ with an increasing possibility as the value increases [32, 38].

**Hazard index (HI).**   From literature survey, it has been recognised that the specific health risk of the heavy metals due to consumption of contaminated vegetable are accumulative and denoted as hazard index (HI) [19, 20, 39]. Consequently, the HI of individual metals sought in this study were calculated using Eq (4).

$$\text{HI} = \sum_{n=1}^{i} THQ_n; i = 1, 2, 3, \ldots, n \tag{4}$$

where HI is hazard index and THQ is the target hazard quotient due to the intake of individual metals through vegetable consumption. It has been pointed out that, when the HI value obtained is less than one, there is no seeming health effect through the exposure to the heavy metals sought. Nevertheless, HI value of greater than one designates probable health effect implication while a serious chronic health impact has been suggested for HI greater than 10.0 [19, 36, 39].

**The target cancer risk (TCR).**   The cancer risk (CR) to the population owing to the intake of specific potentially cancer causing metals were appraised by employing Eq (5) following the same procedure we have recently used [19]. At the same time, the target cancer risk (TCR) ensuing from the ingestion of heavy metals such as As, Pb, Cd, Cr and Ni were estimated by employing Eq (6) as defined by Kamunda and co-investigators [40].

$$CR = EDI \text{ x } CPSo \tag{5}$$

$$TCR = \sum_{n=1}^{i} CR; i = 1, 2, 3, \ldots, n \tag{6}$$

where $CR$ denotes cancer risk over lifetime due to specific heavy metal intake, $EDI$ = estimated daily metal ingestion of the populace in mg/day/kg body weight, $CPSo$ = oral cancer slope factor in $(\text{mg/kg/day})^{-1}$ and n = number of heavy metals considered for cancer risk calculation. The $CPSo$ values used for CR calculation were 1.7 for Ni [41], 0.5 for Cr [42], 1.5 for As [38], 0.0085 for Pb [43] and 0.38 for Cd [44] in $(\text{mg/kg/day})^{-1}$.

**Table 1. Physicochemical characteristics of soils samples from Koka area farmlands in Ethiopia.**

| Physicochemical parameters | | Koka Ejersa area | | Koka Negewo area | |
|---|---|---|---|---|---|
| | | area under tomato cultivation | area under cabbage cultivation | area under tomato cultivation | area under cabbage cultivation |
| pH (1:2.5) | | 7.82±0.02 | 7.68±0.01 | 7.71±0.01 | 7.80±0.02 |
| EC in µS/cm | | 686.33±1.15 | 694.12±2.46 | 828.66±2.08 | 831.01±1.97 |
| %OC | | 1.34±0.02 | 1.41±0.03 | 1.200±0.01 | 1.09±0.02 |
| %OM | | 2.30±0.03 | 2.28±0.02 | 2.06±0.01 | 2.04±0.02 |
| %MC | | 19.65±0.01 | 19.92±0.04 | 24.53±0.01 | 25.24±0.03 |
| CEC in (cmol (+) /kg | | 38.24 ±0.20 | 37.98±0.46 | 41.44±0.38 | 42.09±0.23 |
| Soil Texture | % clay | 48.41±0.61 | 49.01±1.02 | 39.16±0.72 | 38.94±0.46 |
| | % silt | 16.66±0.38 | 17.12±0.65 | 22.50±1.08 | 23.04±0.92 |
| | % sand | 35.25±0.43 | 36.02±1.02 | 38.91±1.01 | 39.19±0.99 |
| Soil class | | Clay | Clay | Clay | Clay |

## Results and discussion

### Soil physicochemical properties

The physicochemical parameters of soil samples on which both vegetables (cabbage and tomato) have been grown in Koka area farmlands were determined and the data is presented in Table 1. The soil texture analysis has revealed that the soil samples from the study farmlands have had a soil texture of clay with clay, silt and sand compositions changing in the assortment of 38.94–48.41, 16.66–23.04 and 35.25–39.19%, respectively. The soil samples examined were observed to be neutral in nature with pH ranged from 7.68 to 7.82. The difference between the pH values of the soil samples were observed to be statistically significant at 95% probability level ($p < 0.05$).

The soil electrical conductivity (EC) of soil samples from Koka Ejersa area were ranged from 686.33–694.12 µS/cm, while the corresponding values for soil samples from Koka Negewo area were ranged between 828.66–831.01 µS/cm. The EC of the soil samples from Koka Ejersa area were statistically significantly different from the corresponding EC values of soil samples from Koka Negewo area farmlands at $p < 0.05$. The EC values found from this investigation were considerably lower than values we have recently reported [19], however, much greater compared with what has been reported by Alghobar and Suresha [45]. The elevated EC amounts found in the soil could be related with the soil texture, as clay texture is anticipated to possess higher EC which associates sturdily to soil particle size and suggesting higher mineral contents of the soil.

The moisture contents (MC) of soil samples from both Koka Ejersa and Koka Negewo areas have ranged between 19.65 to 25.24%, while the percentage organic matter (OM) was in the range of 2.04 to 2.30%. The % MC values of the soil samples were statistically varied meaningfully from each other at 95% confidence level ($p < 0.05$) in correspondence to the sampling location. The % OM of soil samples considered in this is found to be very comparable with what have been reported by Sharma and co-workers [46]. However, a much higher percentage organic carbon (44.9%) have been reported by Balkhair [47] for soil samples from Saudi Arabia. From the comparatively subordinate % OM found in this study, it can be articulated that there has been an extreme cultivation and soil loss in the area.

The soil samples analyzed have shown a cation exchange capacity (CEC) ranging from 37.98 to 42.09 cmol (+) /kg. The CEC value of a soil provides an understanding about the productiveness and nutrient retaining capacity of soil as explicated by Mukhopadhyay and co-researchers [48]. The high CEC values obtained from this study could signify clay texture of

the soil in association with the organic matter embrace electrically charged sites which has the capacity to evoke and grip conversely charged ions as enlightened by Mukhopadhyay and co-researchers [48].

## Heavy metal contents in soil and vegetable samples

**Heavy metal levels in soil samples.** The concentrations of heavy metals in soil samples collected from Koka Ejersa and Koka Negewo in central Ethiopia were determined and the data is as presented in Table 2. Arsenic (As) was detected in all soil samples with mean concentration ranging from 20.90 mg/kg in soil samples from Koka Negewo area on which tomato was cultivated to 31.42 mg/kg in soil samples on which cabbage was cultivated at Koka Ejersa. The concentration of As in the soil samples from the two areas were observed to exceed the acceptable limit of 20 mg/kg set by European Union and 15 mg/kg set by Japan [49]. The toxicity of arsenic (As) to the living being has been well documented despite its continued use as evidenced from the literature [50]. The elevated amounts of As found in the soil samples examined in the current investigation is a witness that this highly toxic metal is still being in use by the in industries positioned around the study location. The arsenic levels in the soil samples from the two locations were observed to statistically significantly differ at 95% probability level.

Higher amounts of lead (Pb) (ranging from 37.30 to 48.60 mg/kg) were found in the soil samples originated from the two locations considered in the investigation. It was observed that the Pb concentrations found in the current study were observed to exceed by about four times the soil standard value of 10 mg/kg as can be seen from Table 2. However, the Pb content of soil samples investigated were found significantly lesser than Indian standard (250–500 mg/kg) as presented by Alghobar and Suresha [45]. Similarly, the levels of Cadmium (Cd) obtained in soil samples were higher than the soil reference value of 0.3 mg/kg with the mean values ranging from 4.36 mg/kg in soil samples on which tomato were cultivated at Koka Negowo to 6.43 mg/kg in soil samples on which cabbage was cultivated at Koka Ejersa. The Cd

**Table 2. Heavy metals concentration (mg/kg) in soil samples analyzed.**

| Metals | amounts of heavy metals (mg/kg) | | | | Soil reference value (mg/kg) |
|---|---|---|---|---|---|
| | Koka Ejersa area | | Koka Negewo area | | |
| | Under Tomato Cultivation | Under Cabbage Cultivation | Under Tomato Cultivation | Under Cabbage Cultivation | |
| As | 27.67±0.05 | 31.42±0.40 | 20.90±0.17 | 29.76±0.23 | 14[a] |
| Pb | 43.60±0.9 | 48.60±0.4 | 37.30±0.00 | 47.2±0.43 | 10[b] |
| Cd | 6.03±0.0 | 6.43±0.37 | 4.36±0.2 | 6.03±0.11 | $\leq 0.3$[c] |
| Zn | 108.26±1.3 | 135.90±0.36 | 97.76±0.64 | 126.76±0.85 | 50[b] |
| Cu | 26.33±0.1 | 28.66±0.49 | 19.83±0.2 | 24.00±0.20 | 20[b] |
| Fe | 52760.00±199.24 | 53806.67±205.26 | 38430.00±418.68 | 49706.67±488.50 | - |
| Mn | 1373.33±20.81 | 1420.00±40.00 | 1500.00±20.00 | 1756.67±40.41 | 2000[d] |
| Cr | 48.10±0.2 | 49.90±1.65 | 49.16±0.3 | 60.73±1.00 | 100[b] |
| Hg | 8.23±0.15 | 7.73±0.11 | 6.13±0.05 | 6.67±0.47 | $\leq 0.3$[c] |
| Ni | 35.00±0.20 | 42.53±0.37 | 40.33±0.45 | 50.73±0.83 | 50[d] |
| Co | 13.53±0.05 | 15.90±0.00 | 18.93±0.30 | 21.70±0.50 | 8[b] |

[a][51];

[b][46];

[c][52];

[d][53]

concentrations found from this investigation were also much higher than what has been quantified by Sharma and co-invesitigators [46]. Zinc (Zn) were detected in all soil samples with concentrations about four times higher than the soil reference value as can be seen from Table 2. Likewise, copper (Cu) was detected in elevated amount in all soil samples obtained from both Koka Ejersa and Koka Negewo farmland with values ranging from 19.83 to 28.66 mg/kg.

The amounts of iron (Fe) detected in this study were observed to be higher in all samples with a minimum of 38430 mg/kg in tomato growing soil sample from Koka Negewo and a maximum of 53806 mg/kg is cabbage growing soil samples from Koka Ejersa. From the result, it can be clearly seen that the farmlands around the Koka area in central Ethiopia are loaded with high levels of minerals. From literature report, we came to learn that a much higher levels of iron (80000 mg/kg) have been reported by McGrath and co-researchers [54]. In contrary, a much lower levels of Fe (11.3 to 62.2 mg/kg) compared with the report of this study were described by Rattan and co-workers [55]. The concentrations of manganese (Mn) in soil samples from the two locations were ranged from 1373 mg/kg in soil samples from Koka Ejersa on which tomato was cultivated to 1756 mg/kg in soil samples from Koka Negewo on which cabbage was cultivated. These concentrations were found to be fairly below the soil standard value of 2000 mg/kg as described by Mahmood and Malik [53].

Mercury (Hg) has been regarded as extremely toxic metal to humans if exposed to it [56]. From the result of this study, it can be seen that significantly elevated concentration of Hg was obtained in soil samples examined with mean concentration ranging from 6.13 mg/kg in tomato growing soil sample from Koka Negewo area to 8.23 mg/kg in the same vegetable growing soil sample from Koka Ejersa area. The amounts of Hg obtained in this study are significantly elevated compared with the maximum limit of 0.3 mg/kg (Table 2). Similarly, an elevated amounts of cobalt (Co) were detected in all soil samples as compared with the soil standard value of 8 mg/kg described by Sharma and co-reserachers [46]. Chromium (Cr) and nickel (Ni) metals were also detected in all the soil samples investigated and their concentrations were ranged from 48.10 to 60.73 mg/kg for Cr and 35.00 to 50.73 mg/kg for Ni. Nevertheless, determined amounts of both Cr and Ni were lower than the soil standard values of 100 and 50 mg/kg, respectively, as indicated on the report by Sharma and co-reserachers [46] and Mahmood and Malik [53]. Generally, the data obtained from this study has clearly shown that, the soil samples from both locations considered for the investigation are evidently polluted by elevated levels of toxic heavy metals.

**Heavy metals contents of vegetable samples.** Heavy metal concentrations in cabbage and tomato samples collected from Koka Ejersa and Koka Negewo area farmlands were examined and the data obtained is presented in Table 3. From the data obtained, it has been witnessed that all the investigated vegetable samples were tested positive to the heavy metals measured. Accordingly, the average concentrations of arsenic (As) have ranged from 0.93 mg/kg (dry weight) in tomato samples from Koka Ejersa to 6.76 mg/kg (dry weight) in cabbage sample from Koka Negewo. The data clearly shows that an elevated level of As were noticed in the vegetable samples regardless of the area of their origin when equated with recommended value of 0.1 mg/kg. It is worth mentioning that the levels of As in cabbage samples are much higher than the values detected in tomato samples. This shows that alike to the data we have reported recently [19], leafy vegetables amass As in substantial quantity than non-leafy vegetables.

Significantly elevated levels of lead (Pb) and cadmium (Cd) were also detected in both cabbage and tomato samples investigated in this study having amounts ranging from 1.8 to 7.26 mg/kg for Pb and 0.33 to 1.03 mg/kg for Cd. Both metals have been observed to contain relatively higher concentration compared with the recommended values indicated in Table 3. The

Table 3. Heavy metal concentrations (mg/kg, dry weight) in vegetable samples analyzed.

| Metals | Concentrations of metals (mg/kg) | | | | Recommended values (mg/kg) |
|---|---|---|---|---|---|
| | Koka Ejersa area | | Koka Negewo area | | |
| | Tomato | Cabbage | Tomato | Cabbage | |
| As | 0.93±0.15 | 6.36±0.28 | 1.03±0.32 | 6.76±0.15 | 0.1[a] |
| Pb | 1.80±0.10 | 6.70±0.26 | 2.63±0.11 | 7.26±0.05 | 0.1–0.3[ab] |
| Cd | 0.33±0.05 | 1.00±0.10 | 0.50±0.00 | 1.03±0.11 | 0.05–0.2[ab] |
| Zn | 17.86±0.28 | 34.4±1.43 | 18.16±0.25 | 36.26±0.35 | 50[c] |
| Cu | 10.70±0.10 | 15.96±0.11 | 10.90±0.26 | 14.16±0.15 | 10–40[ab] |
| Fe | 56.06±0.50 | 336.40±1.25 | 70.66±2.98 | 455.60±2.36 | - |
| Mn | 16.9±0.10 | 102.5±2.40 | 19.06±0.15 | 81.86±0.70 | 500[c] |
| Cr | 0.86±0.05 | 4.50±0.10 | 0.90±0.10 | 5.16±0.05 | 1–2.3[ac] |
| Hg | 3.23±0.05 | 4.36±0.11 | 3.26±0.05 | 4.16±0.05 | 0.01–0.3[bd] |
| Ni | 0.90±0.10 | 3.06±0.15 | 1.33±0.05 | 2.86±0.05 | 10[a] |
| Co | 0.33±0.05 | 1.33±0.05 | 0.46±0.05 | 1.36±0.05 | 50[c] |

[a] [7];

[b] [57];

[c] European union standards [53];

[d] Dutch target value [58]

levels of these toxic metals have showed significant difference in respective to the type of vegetable sample at 95% probability levels ($P < 0.05$). Similar to As, both Pb and Cd were observed to significantly accumulate in leafy vegetable than the fruity vegetables counterparts. Compared with the data obtained from this study, a relatively lower amounts of Pb, Cd and As were reported by Chen and co-investigators [37] in tomato and cabbage samples from China.

The concentrations of other metals including Zn, Cu, Fe, Mn, Ni and Co found in both cabbage and tomato were comparatively less than the recommended values as can be seen from Table 3. The obtained amounts of the metals were ranged from 17.86 to 36.26 mg/kg for Zn; 10.70 to 15.96 mg/kg for Cu; 56.06 to 455.6 mg/kg for Fe; 16.9 to 102.5 mg/kg for Mn; 0.9 to 3.06 mg/kg for Ni and 0.33 to 1.36 mg/kg for Co. The amounts of Zn, Cu, Fe, Mn, Ni and Co in both vegetables were statistically significantly differ from each other at 95% confidence level ($p < 0.05$) regardless of sample locations. This again clearly magnify that leafy vegetables are potentially accumulating higher levels of heavy metals compared with fruity vegetables. Chromium (Cr) and Mercury (Hg) on the other hands have been obtained in all the vegetable samples regardless of the sample's origin. However, the levels of chromium in cabbage were exceeded recommended values, while the corresponding levels in tomato samples were less than the reference values. On the other hand, it has been witnessed that the levels of Hg obtained in all vegetable samples were dangerously exceeded the recommended value 0.01 mg/kg with its values ranging from 3.23 to 4.36 mg/kg. this clearly solidify that the vegetables being grown in Koka area farmlands are dangerously contaminated with an elevated amount of the highly toxic metal, Mercury.

## Bioconcentration factor (BCF)

It has been regarded that transfer and deposition of toxic metals from soil to plant is the major path to the admission of possibly poisonous metals into the food chain [59]. Sharma and co-investigators [46] have pointed out that the speed of transference and buildup of the toxic

**Table 4. Toxic metals bioconcentration factor (BFC) for cabbage and tomato samples.**

| Metals | Bioconcentration factor | | | |
| --- | --- | --- | --- | --- |
| | Koka Ejersa area | | Koka Negewo area | |
| | Tomato | Cabbage | Tomato | Cabbage |
| As | 0.034 | 0.002 | 0.049 | 0.227 |
| Pb | 0.041 | 0.138 | 0.071 | 0.154 |
| Cd | 0.055 | 0.156 | 0.115 | 0.171 |
| Zn | 0.165 | 0.253 | 0.186 | 0.286 |
| Cu | 0.406 | 0.557 | 0.550 | 0.590 |
| Fe | 0.001 | 0.006 | 0.002 | 0.009 |
| Mn | 0.012 | 0.072 | 0.013 | 0.047 |
| Cr | 0.018 | 0.090 | 0.018 | 0.085 |
| Hg | 0.392 | 0.564 | 0.532 | 0.624 |
| Ni | 0.026 | 0.048 | 0.033 | 0.056 |
| Co | 0.024 | 0.084 | 0.024 | 0.063 |

metals to plants differ mainly based on aspects including types of plant types, level and kinds of toxic metals, physical and chemical behaviors of the soil itself among other.

To evaluate the transferability of the metals considered in this study from soil to vegetable samples, the bioconcentration factor (BFC) have been calculated and the data is depicted in Table 4. It can be seen from the data that Cu has shown a higher transfer factor of 0.406 in tomato sample from Koka Ejersa area followed by Hg with BCF of 0.392. Likewise, Cu has shown higher transfer capability in tomato from Koka Negewo as well with BCF = 0.550. However, Hg has showed higher transfer capability in cabbage samples from both Koka Ejersa and Koka Negewo area farmlands with BCF values of 0.564 and 0.624, respectively.

From the results of the current investigation, cabbage has been witnessed to accrue Hg to larger amount, while tomato accumulated Cu. Generally, even if the bioconcentration factor data attained from this particular investigation were all less than one, it was noticed that cabbage had accrued toxic metals to larger degree when arbitrated in comparisons with tomato.

## Heavy metals health risk assessments

**The estimated daily intake (EDI).** The average levels of individual metals in apiece vegetable from the two sampling locations, the estimated daily intake (EDI) of metals by adult population were estimated by making use of Eq (2). The result obtained together with the maximum tolerable daily ingestion (MTDI) for individually metal is depicted in Table 5.

The EDI of heavy metals for an adult person because of the ingestion of 240 g/day of tomato were observed to be $3.08 \times 10^{-4}$, $6.95 \times 10^{-4}$, $1.30 \times 10^{-3}$, $2.76 \times 10^{-4}$, $1.02 \times 10^{-3}$, $3.50 \times 10^{-4}$ and $1.24 \times 10^{-4}$ mg/day/kg body weight for As, Pb, Cd, Cr, Hg, Ni and Co, respectively, while the respective EDI values from consumption of same amount of cabbage were $2.06 \times 10^{-3}$, $2.19 \times 10^{-3}$, $3.19 \times 10^{-4}$, $1.52 \times 10^{-3}$, $1.18 \times 10^{-3}$, $9.29 \times 10^{-4}$ and $4.22 \times 10^{-4}$ mg/day/kg body weight, respectively. The EDI of toxic metals calculated were observed to be less than the MTDI as can witnessed from data in Table 5.

The total estimated daily intake of all metals by an adult person from the ingestion of tomato was 0.038 mg/day/kg body weight, whereas the corresponding amount from the ingestion of cabbage was 0.177 mg/day/kg body weight. The estimated daily intake of arsenic, cadmium, lead and zinc found from this investigation were higher than the numbers described by Shaheen and co-investigators [7] for the same vegetable ingestion.

**Table 5. The EDI of heavy metals in mg/day/kg body weight from consumption of polluted cabbage and tomato.**

| Metals | EDI Values (mg/day/kg body weight) | | Total EDI through consumption of both tomato and cabbage | Maximum acceptable Daily Intake (MTDI) (mg/day) |
|---|---|---|---|---|
| | Tomato | Cabbage | | |
| As | 3.08E-04 | 2.06E-03 | 2.37E-03 | 0.13[a] |
| Pb | 6.95E-04 | 2.19E-03 | 2.89E-03 | 0.21[a] |
| Cd | 1.30E-04 | 3.19E-04 | 4.49E-04 | 0.02–0.07[abc] |
| Zn | 5.65E-03 | 1.11E-02 | 1.67E-02 | 60–65[ab] |
| Cu | 3.39E-03 | 3.63E-03 | 7.02E-03 | 2.5–3[bc] |
| Fe | 1.99E-02 | 1.24E-01 | 1.44E-01 | 15[c] |
| Mn | 5.64E-03 | 2.89E-02 | 3.46E-02 | 2–5[ac] |
| Cr | 2.76E-04 | 1.52E-03 | 1.79E-03 | 0.035–0.2[ac] |
| Hg | 1.02E-03 | 1.18E-03 | 2.20E-03 | 0.04[b] |
| Ni | 3.50E-04 | 9.29E-04 | 1.28E-03 | 0.1–0.3[ac] |
| Co | 1.24E-04 | 4.22E-04 | 5.46E-04 | 0.05[c] |
| **Total** | **3.75E-02** | **1.77E-01** | **2.14E-01** | |

[a][7];

[b][60];

[c][61]

**The target hazard quotient (THQ) and target cancer risk (TCR).** The noncancer causing human health risk from ingestion of vegetables polluted by toxic metals were assayed through the calculation of target hazard quotient (THQ) and the data obtained is presented in Table 6. On the other hand, the target cancer risk because of the acquaintance with heavy metals including arsenic, lead, cadmium and nickel were apprised by making use of heavy metals estimated daily intake values and oral cancer slope factor (CSPo) of each metal.

**Table 6. THQ and TCR to toxic metals from ingestion of polluted vegetables (tomato and cabbage) originated from Koka Ejersa and Koka Negewo areas.**

| Metals | THQ[a] | | TDHQ[c] | TCR[b] | |
|---|---|---|---|---|---|
| | Tomato | Cabbage | | Tomato | Cabbage |
| As | **1.025** | **6.863** | **7.888** | **4.39E-04** | **3.00E-03** |
| Pb | 0.199 | 0.626 | 0.825 | 4.81E-06 | 1.79E-05 |
| Cd | 0.130 | 0.319 | 0.449 | 3.94E-05 | **1.19E-04** |
| Zn | 0.019 | 0.037 | 0.056 | - | - |
| Cu | 0.085 | 0.091 | 0.175 | - | - |
| Fe | 0.028 | 0.178 | 0.206 | - | - |
| Mn | 0.040 | 0.207 | 0.247 | - | - |
| Cr | 0.092 | 0.505 | 0.597 | **1.35E-04** | **7.08E-04** |
| Hg | **3.395** | **3.934** | **7.328** | - | - |
| Ni | 0.017 | 0.046 | 0.064 | **4.81E-04** | **1.64E-03** |
| Co | 0.413 | **1.407** | **1.820** | - | - |
| **HI[d]** | 5.444 | 14.211 | 19.655 | | |

[a] values indicated in bold have shown THQ > 1

[b] values indicated in bold have exceeded the upper limit (1 x 10$^{-4}$) for acceptable risk of developing cancer

[c] TDHQ = the sum of individual metals THQ for both vegetables

[d] HI = Hazard Index

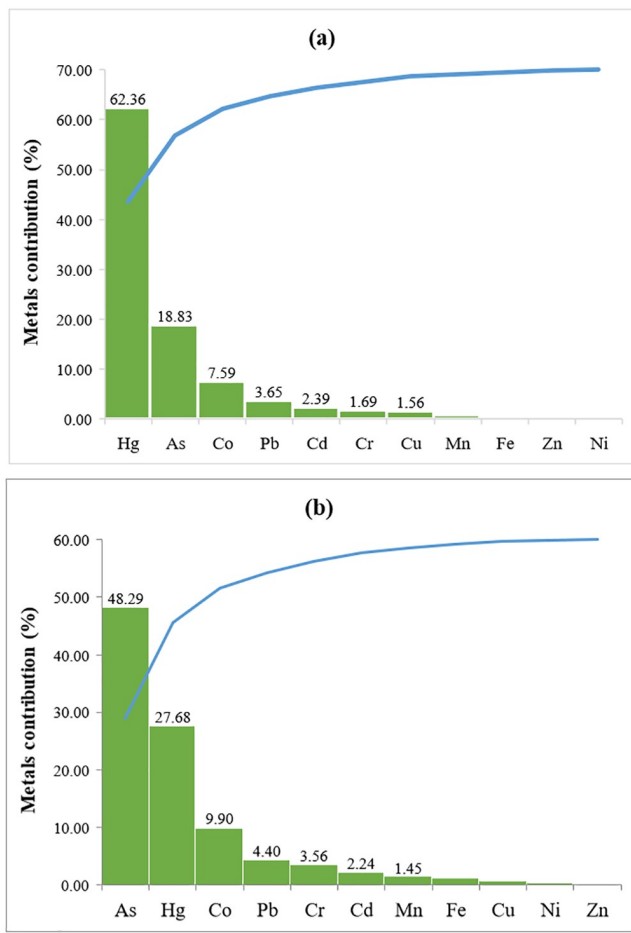

**Fig 2. Individual metals contribution (%) to the health index (HI) from the ingestion of tomato (a) and cabbage (b).**

The THQs for arsenic and mercury in tomato sample were > 1 with values of 1.025 and 3.395 for As and Hg, respectively. This is a clear signal that the ingestion of tomato originated from both Koka Ejersa and Koka Negewo areas could instigate a health risk to the public in the area and beyond. Likewise, the THQ values for arsenic, mercury and cobalt were also > 1 due to cabbage ingestion. As can be observed from the data in Table 6, the TCR value as a result of revelation to toxic metals like arsenic, chromium and nickel through the consumption of tomato were greater than the maximum threshold value of 0.0001. This clearly signifies the high risk of cancer to the adult people from the ingestion of tomato. Likewise, the TCR values of arsenic, cadmium, chromium and nickel also found to exceed the threshold value of 0.0001 due to the ingestion of cabbage originated from the study areas and indicating the possible cancer hazard to the populace in the area and beyond.

**Hazard index (HI).** The collective effect from the absorption of highly toxic metals from the ingestion of various vegetables were estimated through the calculation of hazard index (HI) as indicated in Table 6. Fig 2 dipicts individual metal contribution to the health index from the consumption of each individual vegetables.

Generally, in agreement with the fact that leafy vegetables accrue toxic metals to larger degree than non-leafy vegetables, nearly 72% of the health index (HI) was associated with cabbage ingestion, whereas tomato ingestion was observed to be responsible for only about 28%.

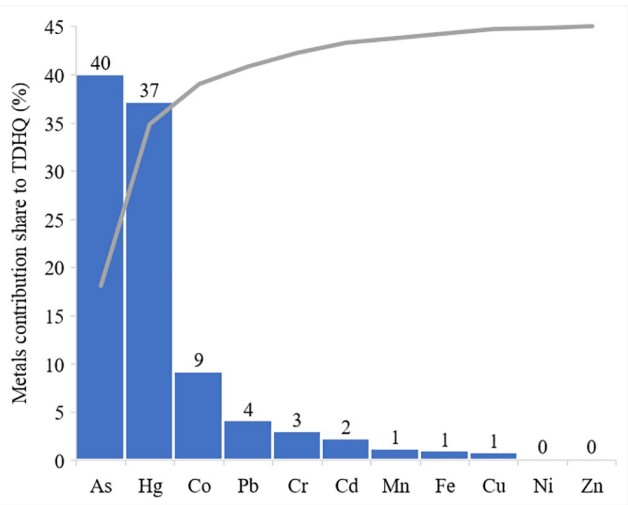

**Fig 3. Contribution share (%) of individual metals to the sum of individual metals THQ (TDHQ).**

Similar to our recent report [19], it has been found out that the major contributors to the total health index (HI) were arsenic and mercury metals. As can be evident from Fig 3, arsenic and mercury have contributed about 40% and 38%, respectively, followed by Co, Pb and Cr which have contributed 9, 4 and 3%, respectively, to the sum of individual metals THQ (TDHQ).

It is important to note that the estimation of EDI, THQ and HI data were based on the projected day-to-day ingestion of vegetables (240 gram per day) for both vegetables and as a result there is a probability that the EDI and THQ values obtained are overestimated. Similarly, only cabbage and tomato were considered in this investigation for the valuation of likely noncancer causing and cancer-causing health hazards to the populace in Koka area. This means that only portion but not the entire hazard to the populace were considered and therefore, it is likely that the possible health hazards to the resident from the acquaintance to toxic metals by the ingestion of vegetables maybe misjudged.

## Conclusions

In this study the amounts of toxic metals in soil and vegetable samples from Koka Ejersa and Koka Negewo areas were assessed. From the result obtained the presence of elevated amounts of toxic metals in both soil and vegetable samples have been observed. Dangerously toxic metals such as As, Pb, Cd, Cr and Hg were detected in frighteningly elevated amounts in both investigated tomato and cabbage samples. This is a clear signal that the population around the study area and beyond are at higher risk of possible health implications through the ingestion of these vegetables. From the result obtained, cabbage was observed to accumulate more toxic metals compared with its counterpart tomato. The EDI of toxic metals from ingesting of both vegetables were observed to fairly lower than the maximum tolerable daily intake of each metal. Nevertheless, it was observed that the THQ of toxic metals from the ingestion tomato were > 1 for As and Hg and for As, Hg and Co due to the ingestion of cabbage. From the HI calculation to estimate the collective noncarcinogenic effects of multiple metals, it has been found that the HI values have exceeded one from ingestion of each tomato and cabbage separately, signifying that about 72% of the consequence is accounted for the ingestion of cabbage unaided. The cancer causing effect investigation has also exposed the presence of total cancer risk (TCR) to the population from As, Cr and Ni due to the consumption of both tomato and

cabbage and from Cd due to the consumption of cabbage, as evidenced from the corresponding TCR values of the indicated metals found to surpass the maximum threshold value of 0.0001. Therefore, due attention should be paid to the safety of vegetables and other food crops being grown and distributed from the area to safeguard the wellbeing of population in the area and beyond.

## Supporting information

**S1 File.**
(PDF)

## Acknowledgments

The authors would like to thank Ambo University and Agricultural and Nutritional Research Laboratory of Ethiopian Institute of Agricultural Research for the laboratory facilities.

## Author Contributions

**Conceptualization:** Leta Danno Bayissa, Hailu Reta Gebeyehu.

**Data curation:** Leta Danno Bayissa, Hailu Reta Gebeyehu.

**Formal analysis:** Hailu Reta Gebeyehu.

**Funding acquisition:** Hailu Reta Gebeyehu.

**Investigation:** Hailu Reta Gebeyehu.

**Methodology:** Leta Danno Bayissa, Hailu Reta Gebeyehu.

**Supervision:** Leta Danno Bayissa.

**Validation:** Leta Danno Bayissa.

**Visualization:** Leta Danno Bayissa.

**Writing – original draft:** Leta Danno Bayissa.

**Writing – review & editing:** Leta Danno Bayissa.

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
