## [Decision Letter · Decision Letter 0]

11 May 2021

PONE-D-21-05216

Vegetables contamination by heavy metals and associated health risk to the population in Koka area of central Ethiopia

PLOS ONE

Dear Dr. Bayissa 

Thank you for submitting your manuscript to PLOS ONE. After careful consideration, we feel that it has merit but does not fully meet PLOS ONE’s publication criteria as it currently stands. Therefore, we invite you to submit a revised version of the manuscript that addresses the points raised during the review process.

ACADEMIC EDITOR: 1- Please write complete information when first time use its abbreviated form.

2- Please add some reference or any information in tableted form which explained WHO criteria for critical limits for the heavy metals in vegetables.

3- What were heavy metals concentration in your studied site and its adjacent areas.

4- Please follow the journal format for text headings and reference style.

5- Please improve your article in scientific manner like sentence structure and English mistakes.

6- Reviewer 3 has attached document which must be addressed by the authors.

We look forward to receiving your revised manuscript.

Kind regards,

Saqib Bashir

Academic Editor

PLOS ONE

Additional Editor Comments:

1- Please write complete information when first time use its abbreviated form.

2- Please add some reference or any information in tableted form which explained WHO criteria for critical limits for the heavy metals in vegetables.

3- What were heavy metals concentration in your studied site and its adjacent areas.

4- Please follow the journal format for text headings and reference style.

5- Please improve your article in scientific manner like sentence structure and English mistakes.

^- Reviewer 3 has attached document which must be addressed by the authors.

Journal Requirements:

2. Please ensure that you refer to Figure 2 in your text as, if accepted, production will need this reference to link the reader to the figure.

3. We note that Figure 1 in your submission contain map images which may be copyrighted. All PLOS content is published under the Creative Commons Attribution License (CC BY 4.0), which means that the manuscript, images, and Supporting Information files will be freely available online, and any third party is permitted to access, download, copy, distribute, and use these materials in any way, even commercially, with proper attribution. For these reasons, we cannot publish previously copyrighted maps or satellite images created using proprietary data, such as Google software (Google Maps, Street View, and Earth). For more information, see our copyright guidelines: http://journals.plos.org/plosone/s/licenses-and-copyright.

You may seek permission from the original copyright holder of Figure 1 to publish the content specifically under the CC BY 4.0 license. 

If you are unable to obtain permission from the original copyright holder to publish these figures under the CC BY 4.0 license or if the copyright holder’s requirements are incompatible with the CC BY 4.0 license, please either i) remove the figure or ii) supply a replacement figure that complies with the CC BY 4.0 license. Please check copyright information on all replacement figures and update the figure caption with source information. If applicable, please specify in the figure caption text when a figure is similar but not identical to the original image and is therefore for illustrative purposes only.

Reviewers' comments:

Reviewer's Responses to Questions

**Comments to the Author**

1. Is the manuscript technically sound, and do the data support the conclusions?

Reviewer #1: Yes

Reviewer #2: Yes

Reviewer #3: Yes

2. Has the statistical analysis been performed appropriately and rigorously? 

Reviewer #1: Yes

Reviewer #2: Yes

Reviewer #3: Yes

3. Have the authors made all data underlying the findings in their manuscript fully available?

Reviewer #1: Yes

Reviewer #2: Yes

Reviewer #3: Yes

4. Is the manuscript presented in an intelligible fashion and written in standard English?

Reviewer #1: Yes

Reviewer #2: Yes

Reviewer #3: Yes

5. Review Comments to the Author

Reviewer #1: The article titled “Vegetables contamination by heavy metals and associated health risk to the

population in Koka area of central Ethiopia.” Is worth accepting, as this study is well designed and is comprehensive, covering all the related elements. But there are few things which I suggest are as follows:

Few suggestions should be incorporated as under:

i) Clearly defined the abbreviations used in the manuscript

ii) Line no. 8 correct the word “glob”

iii) Line no. 373 “Figure 1 here”. It will be replaced with the Figure 3 given below on line no. 375.

Over all I have seen some grammatical mistakes and few typing mistakes in the manuscript. As they are treated as accepted.

Reviewer #2: 1)time field studies have very limited validity, was the experiment replicated on other site or many years? I recommend to do experiment for the one more years to publish your results and use two years data to write your manuscript

2) The abstract is too vague and actual data is required

3) There are some major issues that need to be addressed. In brief, the weakest points of the manuscript are language and structure, and significant efforts should be made to improve both of these issues. At many points results and discussion are difficult to understand due to the poor English level. In its current form is it really difficult to follow the manuscript in places.

4) Use international units system

5) Introduction section must be rewrite with more emphasize on objectives and novelty. The research question is not clear throughout the introduction, which also lacks a clear hypothesis.

6) Avoid the use of acronyms and abbreviations in the conclusions section. Remember that this section must be self-explanatory. Please, in this section you must emphasize the novelty and implication of your study and not just repeat results. Conclusions are not a summary.

7)Methods are not clear, how experiments were conducted, what and how were the application modes etc.?

8)What is the main hypothesis of this study?

9)Introduction". The Authors should clearly state the novelty of the research paper. How is the study improving the state of art? What is the novation/impact of the presented technique?

10)Highlights should be shorter, and provide key points.

11)Abstract: more content should be added to abstract, especially on the general conclusion obtained from this study.The abstract should be more concise.

Reviewer #3: The manuscript PONE-D-21-05216 was reviewed properly. Objective of the manuscript were tried to accomplished through material and methides. Data was analyzed properly and interpreted properly. There are some ambiguity which were asked to the authors in the reviewer's uploaded sheet. Conclusions are reflecting the actual figure of the research work.

In Abstract, TCR write completely when write first time and can be abbreviated next time writing.WHO criteria for critical limits (maximum, minimum and threshold) for the heavy metals in cabbage and tomato were not considered. Why?What are the concentrations of the heavy metals in the lake KOKA water due to industries existing near the KOKA lake? Heavy metals concentrations in the KOKA Lake water were not included in the Manuscript. Why?In methodology, total of 2 cabbage and two tomato samples from both field areas were collected. Were these samples enough for manuscript data?In methodology; Page No.4 and Line number 80; convert “steeliness” into “stainless”.Total 4 soil samples from the study site were taken. Only 4 samples are enough for the manuscript data?In “Result and Discussion” Table No. 1. No need to write word “Beneath”. It is better to write “area under tomato cultivation”. Rest of the correction should be incorporated.Page No. 9. Line No. 186; values 2.06 to 2.30 for %MC are not according to the Table No. 1. The values for %MC are 24.53 and 19.65. Formerly mentioned values (2.06 & 2.30) are for %OM according to Table No. 1. Similarly, in the Line No. 186 and 187; values 1.09 and 1.41 are for %OC, not for %OM, so, correct them according to the Table No. 1.In the line No. 190; value for %OC 44.9% seems to be doubted. So, author should check it.Mention effects of heavy metals under discussion on human body like causing different diseases and disorders in the human beings.    Page No. 13, Line No. 281; value 16.96 is not according to its concerned Table No.3.Line No. 282; value for Ni is 09, which is not according to the Table No. 3.English language for the manuscript should be improved.**********

6. PLOS authors have the option to publish the peer review history of their article (what does this mean?). If published, this will include your full peer review and any attached files.

Reviewer #1: **Yes: **Dr. Muhammad Adnan Bukhari

Reviewer #2: No

Reviewer #3: No

---

## [Author Response · Author response to Decision Letter 0]

27 May 2021

General Remark

Above all, we are very much grateful for the commitment of PLOS ONE Editors and staffs. We really appreciate the swift response and genuine & very constructive review comments we have received on our manuscript. We have gone through our manuscript once again and tried to address all the comments forwarded as much as possible. 

Response on the points raised by the Academic Editor

1. I have carefully looked at the abbreviations we have used and defined when it was first mentioned.

2. WHO/FAO based critical limits/recommended values for heavy metals in both soil and vegetable have already been included (cf. Tables 2&3).

3. We have recently reported levels of heavy metals in both soil and vegetables from the nearby areas and stated that this project is the continuation of that. Elevated heavy metals were reported (cited in this manuscript, https://doi.org/10.1371/journal.pone.0227883) as these areas are home for many small-scale industries.

4. We have checked once again the journal format and made corrections accordingly. 

5. We have gone through our manuscript and tried our best to make necessary corrections as per the suggestion.

Response to Reviewers Comment

Reviewer #1

The comments forwarded are very constructive and helped us to enrich our paper.

i) Corrections were made as per the suggestion

ii) We have corrected the comment

iii) We have corrected it.

We have gone through our manuscript once again and tried to correct some grammatical mistakes and type errors.

Reviewer #2

We really appreciate the reviewer for the genuine and constructive comments. The comment given are very helpful and therefore, all the suggestions have been given due attention and corrections have been made accordingly.

1. Correction was made

2. WHO/FAO based critical limits/recommended values for heavy metals in both soil and vegetable have already been included (cf. Tables 2&3).

3. Our focus in this study was to evaluate the levels of heavy metals in the soil and commonly used vegetables and we haven’t considered the investigation of heavy metals in Koka water this time.

4. For each vegetable from each filed (as explained under methodology section 2.2), five different samples were collected and pooled together to obtain a composite sample. That means five cabbage sample from Koka Ejeresa comprising one representative sample after mix and Five similar samples from Koka Negewo area. The same is true for the cabbage samples as well.

5. Line 80 >> corrected accordingly.

6. Soil samples were taken from the exact location where vegetables have been samples. So, five different samples were taken and pooled together to obtain representative sample from each location. 

7. Table 1 >> corrected accordingly

8. That was a mistake and it has been corrected, thank you very much.

9. I agree that the value seems doubted, but we have checked the article once again and it is what has been reported.

10. We have stated the effects of heavy metals on human being under introduction section. As there are ample of scientific articles discussing the same, we have made abstinence on going in detail into it. 

11. Corrected

12. Corrected

13. We have gone through our manuscript and made necessary corrections as per the suggestions.

---

## [Decision Letter · Decision Letter 1]

23 Jun 2021

Vegetables contamination by heavy metals and associated health risk to the population in Koka area of central Ethiopia

PONE-D-21-05216R1

Dear Dr. Bayissa,

We’re pleased to inform you that your manuscript has been judged scientifically suitable for publication and will be formally accepted for publication once it meets all outstanding technical requirements.

Kind regards,

Saqib Bashir

Academic Editor

PLOS ONE

Additional Editor Comments (optional):

Reviewers' comments:

Reviewer's Responses to Questions

**Comments to the Author**

1. If the authors have adequately addressed your comments raised in a previous round of review and you feel that this manuscript is now acceptable for publication, you may indicate that here to bypass the “Comments to the Author” section, enter your conflict of interest statement in the “Confidential to Editor” section, and submit your "Accept" recommendation.

Reviewer #3: All comments have been addressed

2. Is the manuscript technically sound, and do the data support the conclusions?

Reviewer #3: Yes

3. Has the statistical analysis been performed appropriately and rigorously? 

Reviewer #3: Yes

4. Have the authors made all data underlying the findings in their manuscript fully available?

Reviewer #3: Yes

5. Is the manuscript presented in an intelligible fashion and written in standard English?

Reviewer #3: Yes

6. Review Comments to the Author

Reviewer #3: Author incorporated all the amendments instructed by my side. He improved the manuscript. Still language can be improved. Language should be legible to the readers. Technical work should be written in a simple way so that the reader can read it with keen interest.

7. PLOS authors have the option to publish the peer review history of their article (what does this mean?). If published, this will include your full peer review and any attached files.

Reviewer #3: No

---

## [Editor Report · Acceptance letter]

29 Jun 2021

PONE-D-21-05216R1 

Vegetables contamination by heavy metals and associated health risk to the population in Koka area of central Ethiopia 

Dear Dr. Bayissa:

I'm pleased to inform you that your manuscript has been deemed suitable for publication in PLOS ONE. Congratulations! Your manuscript is now with our production department. 

Kind regards, 

on behalf of

Dr. Saqib Bashir 

Academic Editor

PLOS ONE